# Gas-Phase Structure of 3,7,9-tris(trifluoromethylsulfonyl)-3,7,9-triazabicyclo[3.3.1]nonane by GED and Theoretical Calculations

**DOI:** 10.3390/molecules28093933

**Published:** 2023-05-06

**Authors:** Bagrat A. Shainyan, Alexey V. Eroshin, Valeriya A. Mukhina, Sergey A. Shlykov

**Affiliations:** 1A. E. Favorsky Irkutsk Institute of Chemistry, Siberian Division of the Russian Academy of Sciences, 1 Favorsky Street, 664033 Irkutsk, Russia; 2Department of Physical and Colloidal Chemistry, Ivanovo State University of Chemistry and Technology, Sheremetevkiy Ave, 7, 153000 Ivanovo, Russia; alexey.yeroshin@gmail.com (A.V.E.); lera.muhina2011@yandex.ru (V.A.M.)

**Keywords:** tris(triflyl)-3,7,9-triazabicyclo[3.3.1]nonane, gas electron diffraction, conformational analysis, DFT calculations

## Abstract

The molecular structure and conformational and rotational composition of 3,7,9-tris(trifluoromethylsulfonyl)-3,7,9-triazabicyclo[3.3.1]nonane **1** have been investigated by synchronous gas-phase electron diffraction/mass spectrometry GED/MS and theoretical calculations (B3LYP and M06-2X with cc-pVTZ and aug-cc-pVTZ basis sets) and compared to the X-ray structure. All 16 possible conformers and rotamers were calculated, differing by the conformations of the two piperazine rings, orientation of the CF_3_ groups relative to these rings, and non-equivalence of the two wings of the butterfly structure. The optimized geometry of the most stable *1-c-out-2-c-out* conformer coincides with that in the crystal. In contrast to only one conformer determined by X-ray, the GED analysis revealed the presence of five conformers, *1-c-out-2-c-out* (**I**), *1-c-in-2-c-out* (**II**), *1-c-out-2-c-in* (**III**), *1-b-out-2-c-out* (**IV**), *1-c-out-2-b-out* (**V**) in the ratio of **I**:(**II **+** III**):**IV**:**V** = 36(10):42(6):22(10):0(10). The experimental results are better reproduced by calculations performed for 428 K (the temperature of the GED experiment) than for 298 K (standard), and most satisfactorily at the M06-2X/aug-cc-pVTZ level of theory.

## 1. Introduction

Gas-phase electron diffraction (GED) measurements provide researchers with unique information on the structure and conformational preferences of the studied molecules, which are often different from that of the same molecules in solution or in the crystal. Whereas in solution, the structure is affected by intermolecular interactions, leading to the formation of different homo- and hetero-associates, and in the crystal, it is mainly determined by packing effects. GED measurements allow us to analyze the structure that is intrinsic to individual molecules and, in the case of the presence of several conformers, to determine not only the structure of free molecules undistorted by intermolecular interactions but also the conformational composition. Therefore, the correct use of results from GED and X-ray is complementary.

Recently, we synthesized 3,7,9-tris(trifluoromethylsulfonyl)-3,7,9-triazabicyclo[3.3.1]nonane **1** by the reaction of N,N-diallyltriflamide TfN(CH_2_CH = CH_2_)_2_ with triflamide TfNH_2_ (Tf = CF_3_SO_2_) and determined its structure by single crystal X-ray analysis [1] (Figure 1). Here, we present the results of the GED analysis and DFT calculations of all conformers and rotamers of molecule **1** in order to compare the results to those obtained by single-crystal X-ray diffraction to determine the structure and the fraction of each conformer in the gas phase equilibrium. Note that the conformer distribution could also be measured by NMR, but for flexible systems such as molecule **1,** it is possible only at low temperatures, <150 K. We used a combination of the two methods when it was possible (see review [2] and references therein), but it cannot be applied to compound **1,** which is insoluble in Freon, the only suitable solvent at these temperatures.

Molecule **1** has the plane of symmetry passing through the nitrogen atoms and both piperazine moieties in the *chair* conformation. Possible conformers and rotamers differing by the conformations of the two piperazine rings, orientation of the CF_3_ groups with respect to these rings, and taking into account non-equivalence of the ‘left’ and ‘right’ wings of the butterfly structure in Figure 1 are shown in Chart 1, only for several representative structures lying in energy (Δ*E*) not higher than 4 kcal/mol above the most stable conformer **I** (Table 1). Numbers *1* and *2* denote ‘left’ and ‘right’ wings, letters *b* and *c*—*boat* and *chair* conformations of the piperazine rings, and *in* and *out* mean the *inward* and *outward* directionality of the CF_3_ group with respect to these rings. The total number of different conformers and rotamers, resembling “the Dancing Men” of A. Conan Doyle, is 16.


**Chart 1:**




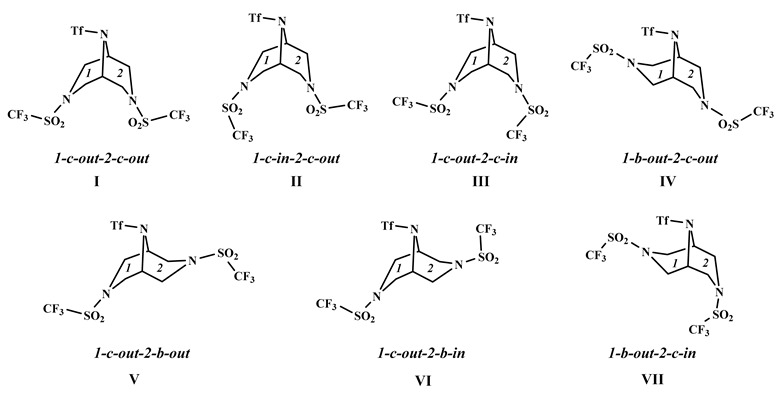



## 2. Results and Discussion

### 2.1. Theoretical Calculations

Table 1 summarizes the total and relative energies, entropies, and relative free energies of all possible conformers and rotamers ranked in the order of rising energy. A rigid bicyclic structure of molecule **1** allows us to neglect other much less stable conformers on the potential energy surface.

In the past, we have used B3LYP or M06-2X in the GED studies of cyclic molecules of similar complexity [3,4,5]. Higher levels of theory require overwhelming computational resources to be practical for such large molecules of low symmetry. The cc-pVTZ basis set provides satisfactory results in a reasonable timeframe. However, diffuse functions may play a role for electronegative atoms F, O, and S; so, we have also performed calculations with the aug-cc-pVTZ basis set. M06-2X functional is more computationally expensive than B3LYP, so the scanning of the triflamide group rotation was performed at the B3LYP/cc-pVTZ level (the rotation barrier of 11 kcal/mol was the same as at M06-2X/cc-pVTZ—*vide infra*).

Note that the calculated geometry of the isolated molecule of the most stable conformer, *1-c-out-2-c-out*, completely coincides with that in the crystal [1]. Expectedly, the *boat* conformers and the conformers with the CF_3_ group directed *inward* the corresponding ring are less stable than their *chair* and *outward* counterparts. The values of Δ*G* vary in parallel with those of Δ*E,* with a few exceptions caused by different flexibility (Δ*S*) of the conformers.

The five most stable structures in Table 1 were recalculated using the aug-cc-pVTZ basis set, as well as applying M06-2X functional with both basis sets. Interestingly, the relative energies Δ*E* follow the same order of stability, whereas for the free energies Δ*G*, this order is violated due to the larger flexibility of the backbone of conformers **II**, **IV**, **V**, with respect to the most stable conformer **I**, see Table 2.

### 2.2. Potential Energy Surface (PES) Profile of Trifluoromethylsulfonyl Groups Rotation

The transition between conformers, in which both cycles have the shape of a chair, can be carried out by rotating one of the trifluoromethylsulfonyl groups. For this reason, the PES profile of the rotation of each of them was scanned around the S–N bonds at the B3LYP/cc-pVTZ level (Figure 2). During the scanning process, the dihedral angle C_β_–N–S–C was changed in increments of 20 degrees.

The rotation barriers of each of the groups were found to be of approx. 11 kcal/mol, which is quite high and can be easily overcome at moderate temperatures: the thermal energy *RT* values are ca. 0.6 and 0.9 kcal/mol at 298 and 428 K, respectively.

Remarkably, the red and blue curves in Figure 2 practically coincide, meaning that the dependence of energy on the rotation of the unequally distant triflyl groups in the two piperazine rings is identical. This may suggest the absence or negligible role of noncovalent interactions in the molecule.

### 2.3. Conformer Contributions vs. Temperature

The synchronous gas-phase electron diffraction/mass spectrometric (GED/MS) experiments for compound **1** were carried out at temperature *T* = 428(5) K, see Experimental section. Since it is different from the standard value of 298 K applied as default in quantum chemical (QC) calculations, we performed a series of scans to estimate the conformers contribution change against temperature, see Computational details section for details.

According to all theoretical approaches applied, the *1-c-out-2-c-out* conformer is predicted to dominate at room temperature in the gas phase (Table 2), as well as in the case of the solid phase [1].

At the same time, as the temperature increases, the relative contribution of the most stable *1-c-out-2-c-out* conformer decreases (Figure 3), although it still predominates up to rather high temperatures, following on from the DFT calculations with B3LYP functional. On the other hand, according to M06-2X/cc-pVTZ calculations, already at ca. 380 K, the *1-c-out-2-b-out* conformer becomes predominant, although all other approaches used indicate its insignificant presence in the vapor. For this reason, it would be incorrect to describe the composition of the vapor by the single *1-c-out-2-b-out* conformer.

### 2.4. Geometries

The internuclear bond distances in the considered conformers are very close to each other, according to QC calculations, as shown in Table 3 below and Appendix A in ESI. Noticeable differences are observed for the distances C_α_C_β_. In the boat-shaped cycle (in the case of conformers *1-b-out-2-c-out* and *1-c-out-2-b-out*), they are ~0.03 Å longer compared to those in the *chair* cycle. Also, in these forms, there is a shortening of the C–N distances. The flap-angle in the *boats* has a lower value than in the *chair*. The average C–F bond lengths are shown in Table 3, although in each of the three trifluoromethyl groups of the molecule, for each of the conformers, the distance C–F is shorter for the atom F lying in the plane of symmetry of the molecule by about 0.01 Å relative to the other two bonds in the CF_3_ group.

At the same time, different combinations of the method/basis set (Appendix A) predict different exocyclic parameters up to 0.04 Å in the values of the internuclear distances S–N and S–C for the same conformer.

The pyramidality of the nitrogen environment in the conformers of **1**, characterized by a sum of bond angles at N-atoms (C–N–C and two C–N–S), is very close to 360°—varying from 354 to 360°. Minimal are the values for **II** and **III** at *inward* orientation of trifluoromethyl groups (see Introduction); maximal—for **IV** and **V**, but in *chairs* only. The experimentally measured X-ray values of Σ_N_ are 358° in **1 [1]**, i.e., the same as in 1,1,1-trifluoromethanesulfonamide [7]. Another parameter of pyramidality/planarity is the angle between the CNC plane and N–S bond, which can be expressed via the S–N–X angle as (180 – S–N–X) and is given in Table 3. Concerning the ‘top’ fragment of the molecule, see line headed ‘SN_t_X’ in Table 3; one can note a maximal extent of its planarity in conformer **IV** (CF_3_ group directed towards the *boat*); however, in contrast, a minimal in **V** (CF_3_ group directed towards the *chair*).

## 3. Structural Analysis

Synchronous GED/MS experiments were carried out at 428(5) K. A ceaseless mass spectrometric monitoring of the vapor under study showed peaks which explicitly allow them to be assigned to the target molecule. The main conditions of the GED/MS experiments and the mass spectra are listed in Table 4 and Table 5, respectively, see Experimental section.

Prior to the refinement of the experimental diffraction intensities, a ‘sensitivity’ of the GED method to the structure of the conformers of **1** was tested by juxtaposition of theoretical molecular scattering sM(s) and radial distribution f(r) functions based on M06-2X/aug-cc-pVTZ calculations. The functions for the *1-c-in-2-c-out* (**II**) conformer were taken as a sort of a reference instead of experimental data, while for those for the other, most stable conformers, the disagreement factors R_f_ (see Equation (S1)) were calculated and the f(r) curves and the differences Δf(r) visualize a possibility to be distinguished, Figure 4.

The theoretically generated conformer structures are clearly distinguishable in terms of GED—the disagreement factors R_f_ are rather high (above 10%). The *1-c-in-2-c-out* and *1-c-out-2-c-in* pairs are exceptions, for which the R_f_ = 2.9 and 4.9% for M06-2X with cc-pVTZ and aug-cc-pVTZ basis sets, respectively. Nevertheless, we may consider these two conformers to noticeably contribute to the gas phase at 428 K because *1-c-in-2-c-out* is expected to be a significant amount. The M06-2X/cc-pVTZ approach predicted similar R-factors.

The UNEX program [8] was used to refine geometric and vibrational parameters using the least-squares method. Vibrational corrections and starting geometries and vibrational amplitudes for the GED data refinement were derived by the VibModule [9] program using a nonlinear relation between Cartesian and internal coordinates on the basis of theoretical calculations.

For all conformers, z-matrices were written with the three imaginary atoms necessary to maintain the stability of the model. In particular, sulfur atoms bound to nitrogen can be described using valence angles S–N–C; however, such a description does not adequately vary the position of the triflamide group relative to the ring, so these atoms were given using imaginary atoms X (angle S–N–X_c_ in the case of a *chair* or S–N–X_b_ in the case of a *boat*). The X_a_ atom is located between the “central” carbon atoms and can explicitly vary the flap angle for the *boat*. The position of imaginary atoms and the symbols of all atoms on the example of the conformer *1-c-out-2-b-out*, which includes both a *boat* and a *chair*, are shown in Figure 5.

The following parameters were used as independent variables in the description of molecular models: a) all r(S–N); b) r(F–C); c) r(C–S); d) r(O–S); e) all r(C–N) and r(C_α_–C_β_); f) r(C–H); g) all a(C–S–N); h) a(O–S–C); i) a(O–S–N_t_); j) a(F–C–S); k) a(C_α_–N_t_–S); l) a(C_β_–C_α_–X_α_); m) a(S–N–X_c_); n) d(C_α_–N_t_–S–C); o) d(C–S–N–X_c_); p) d(N–C_β_–C_β_–C_α_); r) d(C–C_α_–X_α_–N). For conformers containing a piperazine fragment in the boat-shaped fragment, the angle group q) a(S–N–X_b_) was added as an independent parameter. The parameters within one group were described by fixing the differences between them obtained according to the QC calculations.

The vibrational amplitudes were combined in the following six groups according to their internuclear distance to the specific groups at the radial distribution (Figure 6b): 0–1.95, 1.95–2.3, 2.3–3.4, 3.4–4.15, 4.15–5.35, 5.35–11 Å. The ratio of amplitudes within the groups was assumed to be equal to theoretical values. The valence and dihedral angles used to describe the position of hydrogen atoms were fixed.

The order of the refinement was as follows: (1) scale factors k, see Equation (S1); (2) bond distances (a)–(e); (3) bond distances (a)–(e), all valence and dihedral angles; (4) vibrational amplitudes; (5) conformers contribution; (6) all geometric parameters (a)–(q), with the vibrational amplitudes and conformers contribution.

Experimental and theoretical molecular scattering intensities sM(s) (Equation (S2)) and radial distribution functions are plotted in Figure 6a and Figure 6b, respectively, along with the ‘Experiment–Theory’ differences.

### 3.1. Scanning the R-Factor vs. Conformer Contributions

GED data refinement was performed using the vibrational corrections, starting geometries and vibration amplitudes generated on the base of four QC levels, B3LYP, and M06-2X with cc-pVTZ and aug-cc-pVTZ basis sets. Both combinations with B3LYP yielded worse fit compared to the experimental molecular scattering intensities, see Appendix A in ESI.

According to the results of the GED data refinement using the theoretical force field from M06-2X/aug-cc-pVTZ calculations, the gas phase of **1** at 428 K, along with the conformer *1-c-out-2-c-out* (42%), contains conformers *1-c-in-2-c-out*, *1-c-out-2-c-in* (total contribution of 40%), and *1-b-out-2-c-out*, which includes a piperazine cycle in the form of a boat. The disagreement factor was R_f_ = 4.195%, see row headed as ‘X_428 K_ (UNEX)’ in Table 3. At the same time, the contribution of the *1-c-out-2-b-out* conformer is close to zero. However, the errors in determining the composition are quite large, in particular, due to the high correlation between the contributions of *1-c-in-2-c-out* and *1-c-out-2-c-in*.

For this reason, R-factor was scanned to explore its dependence on the composition with a fixed contribution of optimized structures in increments of 10% for each of the conformers (number of points = 1001), while only scale factors varied. Then, similar scans were performed in the minimum area with a smaller step (4%, 1991 point and 2%, 5369 points, see Appendix A). The composition determined in this way was as follows: **I**:(**II** + **III**):**IV**:**V** = 36(10):42(6):22(10):0(10) mol.%. The errors were determined using the Hamilton’s criterion [2] (at a significance level of 0.05) with a ratio R_f_/R_f_(min) = 1.03.

The fact that the system cannot be described by a single conformer can be confirmed by the R_f_ values for a composition comprising 100% of a single conformer. So, the disagreement factor for *1-c-out-2-c-out* was 7.78%, while for the refined composition was only 4.195%, see rows marked as ‘R_f_ (ind.)’ and ‘R_f_ (tot.)’, respectively, in Table 3. At the same time, for geometry taken from QC calculations without refinement of parameters, R_f_ exceeds 14%.

### 3.2. Comparison of Gas-Phase and Crystal Structures

Unlike conformer variety in the gas phase described above, a single conformer was detected by single-crystal study XRD [1], namely *1-c-out-2-c-out* (**I**). Compared with that determined in the solid phase [1], the C–C and C–N distances of the piperazine ring for free molecule **1**, determined from the GED experiment, are significantly shorter (1.518 vs. 1.533/1.535 Å for the C–C and 1.461 vs. 1.479 Å for C–N). On the contrary, in the gas phase, where the intermolecular interactions are negligible, the exocyclic S–N, S–C, and C–F bonds are longer, though in the case of the latter, this is to a smaller extent.

## 4. Conclusions

The structure of free molecule **1** was studied using QC calculations and GED. The GED method provided unique information on the structure and is not distorted by intermolecular interactions, which cannot be obtained by methods such as NMR in solution or X-ray in the crystal. It also allowed us to experimentally determine the conformational distribution and to show that it cannot be described by only one *1-c-out-2-c-out* conformer as in the crystal and that ‘liberated’ molecules behave in a distinctly different way. According to our study, its contribution does not exceed 50%, although it is still most abundant in gas. The refinement of the GED experiment data of **1** was not able to reliably distinguish conformers **II** and **III**, but proved their essential summed contribution. Surprising was a noticeable amount of conformer **IV**, in which one of the rings had the shape of a *boat*. QC calculations confirmed a decrease in the fraction of conformer **I** with increasing temperature, which does not contradict the data for the crystal measured at 100(2) K, i.e., 330 K lower than for gas phase. The endocyclic bonds are shorter in free molecule **1** than those in the crystal, whereas the exocyclic bonds are significantly longer. The M06-2X/aug-cc-pVTZ best fits the experimental GED, R_f_ = 4.2%. Scanning the potential energy surface revealed that the energy barriers for transitions **I → II** and **I → III** by the triflamide group rotation are 11 kcal/mol.

The noticeable sensitivity of GED to such a complex conformational composition, as in the case of compound **1**, is largely due to the presence of heavy fluorine atoms in the terminal CF_3_ groups, which are less flexible than methyl groups. The obtained results allow us to extend the methodology of the present work to our ongoing studies of other conformationally flexible cyclic products.

## 5. Experimental Section

### 5.1. Synthesis

Compound **1** was obtained as a by-product of the reaction of N,N-diallyltriflamide with triflamide in the oxidative system NaI/*t*-BuOCl in MeCN upon cooling and separated by washing out the product mixtures with chloroform and the crystallization of undissolved solid from a minimal amount of chloroform. Colorless crystals, m.p. 283 °C.

### 5.2. Synchronous GED/MS Experiments

Synchronous gas-phase electron diffraction and mass spectrometric GED/MS experiments for 1 were carried out using the combined EMR100/APDM-1 apparatus described earlier [10,11,12,13]. The sample of 1 was loaded into molybdenum effusion cell at atmosphere. Detailed conditions of the GED/MS experiment are listed in Table 4. The temperature of 428 K is the lowest temperature at which the minimal required vapor pressure of 0.05–0.1 Torr could be obtained. The effect of pressure itself can apparently be neglected because, in gas phase, it is manifested only when the number of species in the equilibrium changes, which is not the case for the equilibrium between conformers.

During all stages of the cell heating, the effusing vapor was continuously monitored by cyclic recording mass spectra each ca. 1.5 min. Mass spectrum is given in Table 5.

### 5.3. Computational Details

CREST program with GFN2-xTB method [14,15] was used for the preliminary conformational search.

DFT calculations (hybrid functional B3LYP and Minnesota functional M06-2X [16] combined with correlation consistent triple-zeta basis set cc-pVTZ and its analogue augmented by diffuse functions aug-cc-pVTZ) [17,18,19] were performed with use of Gaussian09 program [20] to obtain the optimized structures and compute the harmonic vibrations. The basis sets were taken from the EMSL BSE library [21].

The influence of the temperature on the conformational preferences in the gas phase of **1** was studied by calculating the Gibbs free energies (ΔG) using VibModule^8^ program at the interval of 100–1000 K with a step of 10 K.

## Figures and Tables

**Figure 1 molecules-28-03933-f001:**
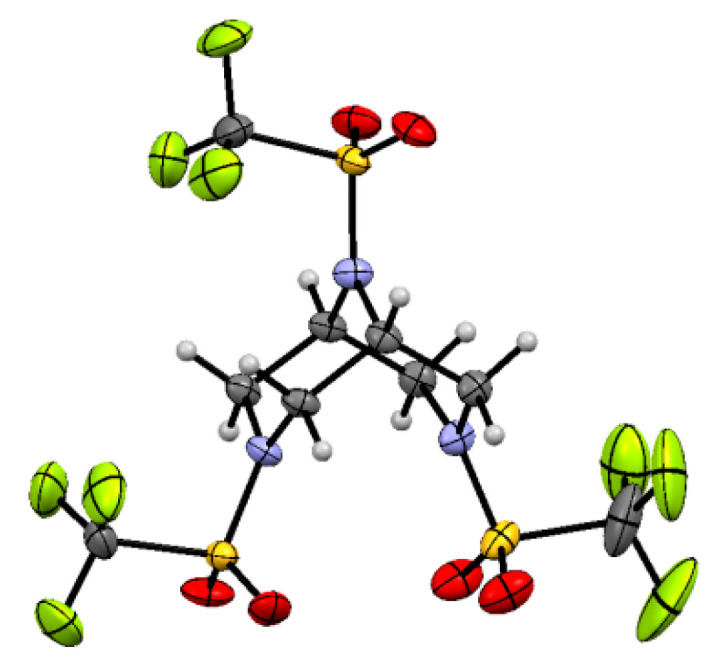
Molecular crystal structure of 3,7,9-tris(trifluoromethylsulfonyl)-3,7,9-triazabicyclo[3.3.1]nonane **1**.

**Figure 2 molecules-28-03933-f002:**
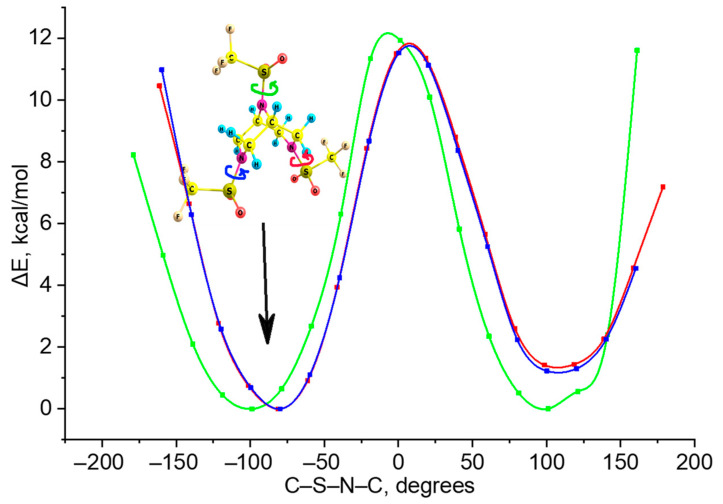
Potential energy surface profiles of the trifluoromethyl groups rotation. The colors of the curves match those of the conformers’ numbering, and arrows indicate the rotations.

**Figure 3 molecules-28-03933-f003:**
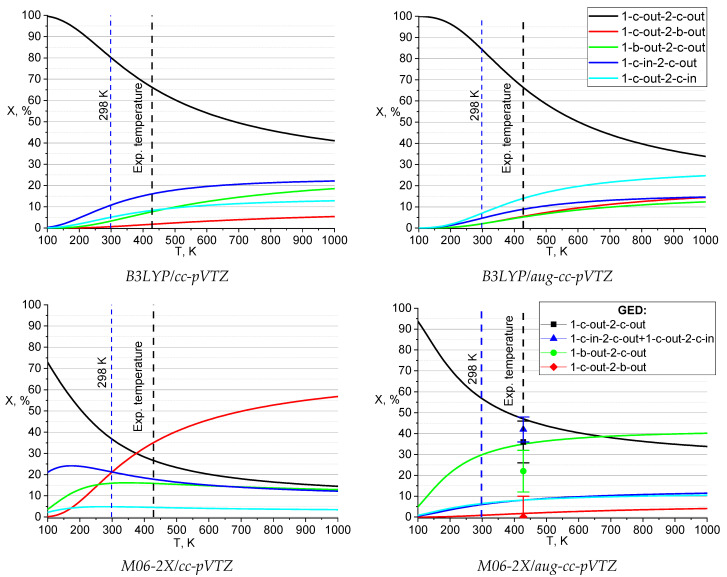
Temperature dependence of conformational composition of **1** from DFT calculations. GED data for 428 K with error limits also given at the M06-2X/aug-cc-pVTZ plot.

**Figure 4 molecules-28-03933-f004:**
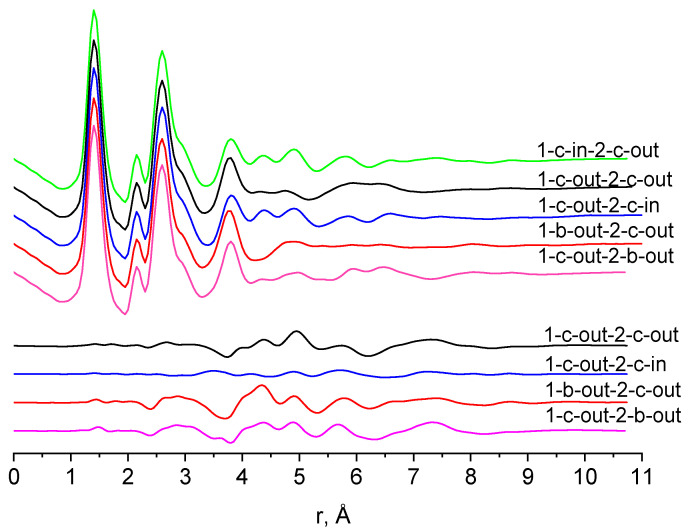
Comparison of theoretical radial distribution functions f(r) of **1**. Difference curves Δf(r) concerning 1-c-in-2-c-out are shown at the bottom. The R-factors are *1-c-out-2-c-out* (13.4%), *1-c-out-2-c-in* (4.9%), *1-b-out-2-c-out* (17.1%), *1-c-out-2-b-out* (14.2%). Molecular parameters were calculated at the M06-2X/aug-cc-pVTZ level.

**Figure 5 molecules-28-03933-f005:**
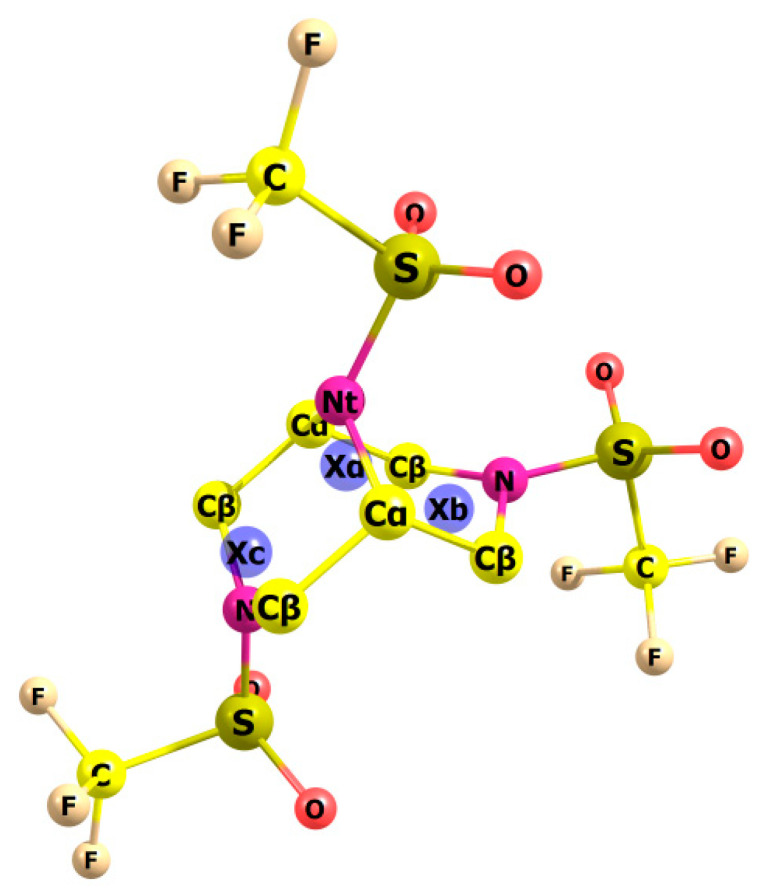
Molecular model of *1-c-out-2-b-out* with dummy atoms with atoms labelling. X_α_ is a dummy atom between two C_α_; X_c_ and X_b_ are dummy atoms between two C_β_ in chair and boat forms of piperazine ring, respectively. Hydrogens have been omitted.

**Figure 6 molecules-28-03933-f006:**
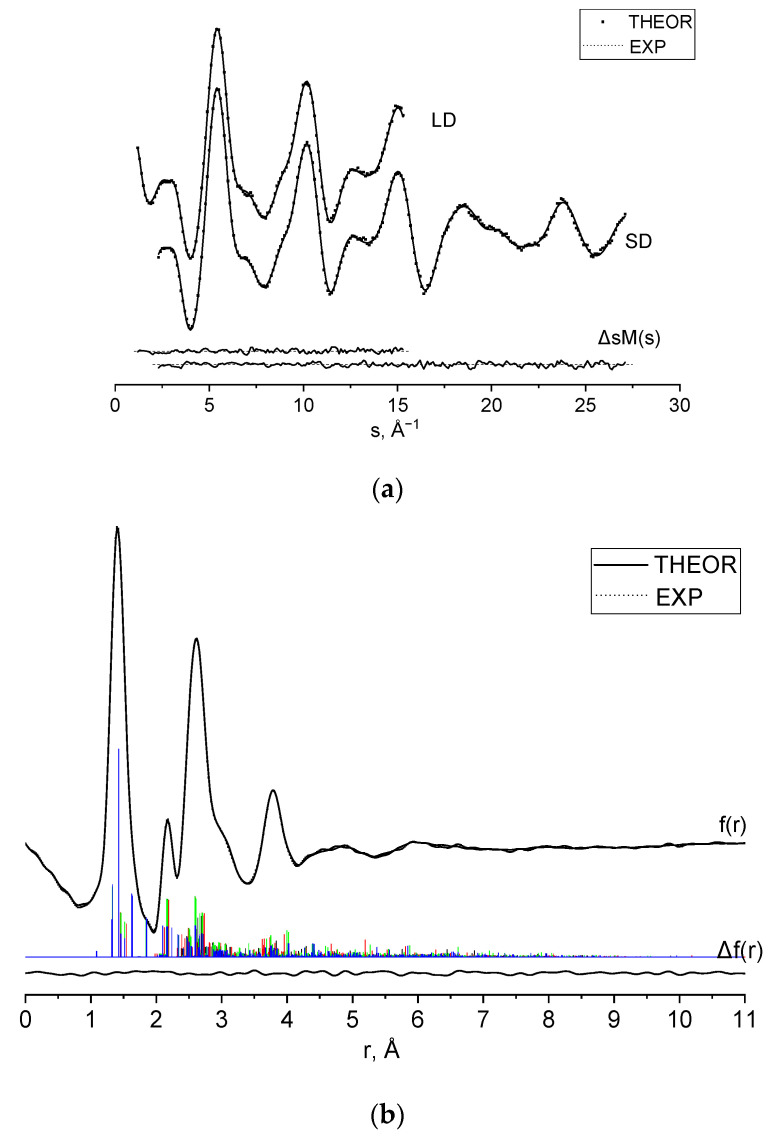
(**a**) Experimental (lines) and theoretical for refined structures and conformer contribution (dots), molecular scattering intensities sM(s), and (**b**) radial distribution curves f(r). Differences “Exp.–Theor.” are given in the bottom. Colored vertical lines describe contribution of conformers to the radial distribution; the colors are the same as those in Figure 4.

**Table 1 molecules-28-03933-t001:** B3LYP/cc-pVTZ total (*E*_tot_) and relative (Δ*E*) energies, entropies (Δ*S*_298_), and relative free energies (Δ*G*_298_) of the calculated structures.

No.	Structure	*E*_tot_ (a.u.)	Δ*E* (kcal/mol)	Δ*S* (e.u.)	Δ*G* (kcal/mol)
**I**	*1-c-out-2-c-out*	−3058.361140	0	0	0
**II**	*1-c-in-2-c-out*	−3058.359208	1.21	−0.11	1.19
**III**	*1-c-out-2-c-in*	−3058.358922	1.39	−1.03	1.64
**IV**	*1-b-out-2-c-out*	−3058.358264	1.80	0.54	1.90
**V**	*1-c-out-2-b-out*	−3058.357618	2.21	−1.62	2.86
**VI**	*1-c-out-2-b-in*	−3058.355866	3.31	3.27	2.49
**VII**	*1-b-out-2-c-in*	−3058.354815	3.97	2.41	3.35
**VIII**	*1-b-in-2-c-out*	−3058.354592	4.11	4.21	3.04
**IX**	*1-c-in-2-b-out*	−3058.354336	4.14	−0.53	4.39
**X**	*1-c-in-2-c-in*	−3058.353321	4.91	−1.17	5.45
**XI**	*1-c-in-2-b-in*	−3058.352995	5.11	−0.60	5.38
**XII**	*1-b-in-2-c-in*	−3058.351577	6.00	−1.12	6.47
**XIII**	*1-b-out-2-b-out*	−3058.345358	9.90	0.82	9.82
**XIV**	*1-b-out-2-b-in*	−3058.344866	10.21	−1.89	10.98
**XV**	*1-b-in-2-b-out*	−3058.343716	10.93	−1.25	11.42
**XVI**	*1-b-in-2-b-in*	−3058.242856	11.47	−2.85	12.42

**Table 2 molecules-28-03933-t002:** Calculated total relative Δ*E* energies, relative free energies Δ*G* (kcal/mol), and mole fraction X (mol.%) of the five most stable conformers of **1** at room, 298 K, and at a temperature of GED/MS experiments, 428 K.

	T, K	Conformer				
		I	II	III	IV	V
**B3LYP/A ^a,c^**						
ΔE		0	1.21	1.39	1.80	2.21
X	298	81	10	5	3	1
ΔG		0	1.19	1.64	1.90	2.86
X*_T_*	428	66	16	8	8	2
ΔG		0	1.20	1.77	1.83	3.06
**B3LYP/B ^c^**						
ΔE		0	1.73	1.87	2.07	2.30
X	298	85	4	7	2	2
ΔG		0	1.72	1.48	2.21	2.22
X	428	66	9	14	5	6
ΔG		0	1.71	1.32	2.16	2.11
**M06-2X/A ^c^**						
ΔE		0	0.48	0.58	0.28	1.49
X	298	37	21	5	16	21
ΔG		0	0.33	1.20	0.50	0.33
X	428	27	18	4	16	35
ΔG		0	0.34	1.51	0.44	-0.24
**M06-2X/B ^c^**						
ΔE		0	1.10	1.02	0.72	1.74
X	298	57	6 ^b^	7	30	1
ΔG		0	1.33	1.29	0.38	2.50
X	428	47	8	8	35	2
ΔG		0	1.49	1.49	0.25	2.81

^a^—A and B are cc-pVTZ and aug-cc-pVTZ basis sets, respectively; ^b^—imaginary frequency was found ν = *i*16 CM^−1^; its sign was put positive at ΔG calculations; ^c^—Cartesian coordinates are listed in ESI.

**Table 3 molecules-28-03933-t003:** Selected geometric parameters ^a,b^ (Å and degrees) along with contributions X (mol. %) of the conformers of **1**.

	*1-c-out-2-c-out*		*1-c-in-2-c-out*		*1-c-out-2-c-in*		*1-b-out-2-c-out*		*1-c-out-2-b-out*	
	I		II		III		IV		V	
	QC	GED ^c^	QC	GED	QC	GED	QC	GED	QC	GED
**Bond distances**										
S–N	1.609	1.623(5)	1.610	1.624(5)	1.610	1.624(5)	1.611	1.625(5)	1.627	1.641(5)
S–C	1.840	1.846(5)	1.840	1.846(5)	1.839	1.846(5)	1.839	1.845(5)	1.840	1.846(5)
C–F (ave.)	1.326	1.326(3)	1.325	1.325(3)	1.325	1.325(3)	1.325	1.325(3)	1.325	1.326(3)
N–C_α_(f)	1.470	1.461(4)	1.468	1.459(4)	1.468	1.459(4)	1.462	1.453(4)	1.467	1.458(4)
N–C_α_(n)	1.470	1.461(4)	1.468	1.459(4)	1.468	1.459(4)	1.462	1.453(4)	1.467	1.458(4)
C_α_–C_β_(1f)	1.527	1.518(5)	1.524	1.515(5)	1.524	1.515(5)	*1.554*	*1.545(5)*	1.525	1.516(5)
C_α_–C_β_(2f)	1.528	1.518(5)	1.527	1.518(5)	1.527	1.518(5)	1.528	1.518(5)	*1.555*	*1.545(5)*
C_α_–C_β_(1n)	1.527	1.518(5)	1.524	1.515(5)	1.528	1.515(5)	*1.554*	*1.545(5)*	1.525	1.516(5)
C_α_–C_β_(2n)	1.528	1.518(5)	1.527	1.518(5)	1.525	1.518(5)	1.528	1.518(5)	*1.555*	*1.545(5)*
C_β_–N(1f)	1.462	1.453(4)	1.467	1.458(4)	1.469	1.460(4)	*1.462*	*1.452(4)*	1.463	1.454(4)
C_β_–N(1n)	1.462	1.453(4)	1.467	1.458(4)	1.470	1.461(4)	*1.462*	*1.452(4)*	1.463	1.454(4)
C_β_–N(2f)	1.467	1.457(4)	1.470	1.461(4)	1.468	1.459(4)	1.465	1.456(4)	*1.461*	*1.452(4)*
C_β_–N(2n)	1.467	1.457(4)	1.470	1.461(4)	1.467	1.458(4)	1.465	1.456(4)	*1.461*	*1.452(4)*
N–S(1)	1.611	1.625(5)	1.622	1.636(5)	1.621	1.635(5)	1.616	1.631(5)	1.610	1.624(5)
N–S(2)	1.616	1.630(5)	1.622	1.636(5)	1.621	1.635(5)	1.611	1.625(5)	1.618	1.632(5)
S–C(1)	1.840	1.847(5)	1.837	1.844(5)	1.846	1.852(5)	1.842	1.848(5)	1.840	1.846(5)
S–C(2)	1.842	1.849(5)	1.846	1.852(5)	1.839	1.845(5)	1.840	1.846(5)	1.842	1.848(5)
S–O (ave.)	1.423	1.425(3)	1.424	1.426(3)	1.424	1.426(3)	1.424	1.426(3)	1.423	1.425(3)
**Bond angles**										
N_t_SC	101.8	99.0(20)	101.7	98.9(20)	101.6	98.8(20)	103.4	103.4 ^f^	99.8	97.0(20)
SN_t_C_α_(f)	122.7	120.4(16)	122.8	120.4(16)	122.9	120.6(16)	123.2	123.2 ^f^	119.5	117.2(16)
SN_t_C_α_(n)	122.7	120.4(16)	122.8	120.4(16)	122.8	120.6(16)	123.2	123.2 ^f^	119.5	117.2(16)
C_α_N_t_C_α_ ^e^	112.6	117.6(36)	112.2	117.2(36)	112.4	117.2(36)	113.5	113.5 ^f^	112.5	117.4(51)
N_t_C_α_C_β_(1f) ^e^	107.8	103.0(10)	108.0	103.2(10)	107.7	102.9(10)	109.8	106.1(6)	107.4	102.6(12)
N_t_C_α_C_β_(2f) ^e^	108.4	103.5(11)	108.4	103.6(11)	108.7	103.9(11)	107.3	103.5(6)	*109.9*	*105.0(14)*
N_t_C_α_C_β_(1n) ^e^	107.8	103.0(10)	108.0	103.2(10)	108.1	102.9(10)	109.8	106.1(6)	107.4	102.6(12)
N_t_C_α_C_β_(2n) ^e^	108.4	103.5(11)	108.4	103.6(11)	108.2	103.9(11)	107.3	103.5(6)	*109.9*	*105.0(14)*
C_α_C_β_N(1f) ^e^	110.2	112.0(13)	111.0	116.0(12)	110.4	117.0(12)	*109.5*	*114.6(15)*	108.7	109.8(14)
C_α_C_β_N(2f) ^e^	110.3	111.8(13)	110.8	116.7(12)	111.0	117.3(12)	108.4	113.1(14)	*109.1*	*110.4(15)*
C_α_C_β_N(1n) ^e^	110.2	112.0(13)	111.0	116.0(12)	111.1	117.0(12)	*109.5*	*114.6(15)*	108.7	109.8(14)
C_α_C_β_N(2n) ^e^	110.3	111.8(13)	110.8	116.7(12)	110.3	117.3(12)	108.4	113.1(14)	*109.1*	*110.4(15)*
C_β_NS(1f) ^e^	121.4	124.1(11)	118.5	121.9(10)	118.6	121.7(10)	*120.4*	*125.5(50)*	122.2	124.6(18)
C_β_NS(2f) ^e^	120.1	123.2(10)	118.6	122.1(10)	119.2	122.8(10)	122.0	126.7(14)	*120.6*	*123.8(18)*
C_β_NS(1n) ^e^	121.4	124.1(11)	118.5	121.9(10)	118.8	121.9(10)	*120.4*	*125.5(50)*	122.2	124.6(18)
C_β_NS(2n) ^e^	120.1	123.2(10)	118.6	122.1(10)	118.1	121.9(10)	122.0	126.7(14)	*120.6*	*123.8(18)*
NSC(1)	102.4	99.6(20)	101.4	98.6(20)	104.3	101.5(20)	*103.1*	*100.3(20)*	103.1	100.3(20)
NSC(2)	103.1	100.3(20)	104.5	101.7(20)	101.4	98.6(20)	103.0	100.2(20)	*102.5*	*99.7(20)*
SN_t_X ^e^	166.9	167.3	166.1	166.4	167.0	167.4	176.4	176.6	152.4	151.5
SNX(1)	172.4	177.9(11)	157.1	162.7(10)	155.4	161.0(10)	*164.1*	*167.5(50)*	176.2	176.2(18)
SNX(2)	163.2	168.8(10)	155.2	160.7(10)	157.4	163.0(10)	176.7	180.0(14)	*164.8*	*170.4(18)*
**Dihedral angles**										
flap(t1) ^d^	57.9	61.1(19)	58.7	61.9(19)	58.4	62.3(19)	*53.2*	*56.4(19)*	58.6	61.7(19)
flap(t2) ^d^	57.2	60.3(19)	57.5	60.6(19)	57.6	60.0(19)	58.5	61.7(19)	*53.6*	*56.8(19)*
flap(1) ^d^	47.1	51.4(14)	42.9	40.2(14)	45.4	38.5(14)	*48.9*	*48.9 ^f^*	51.7	56.7(14)
flap(2) ^d^	46.4	51.4(14)	45.3	40.2(14)	44.2	38.5(14)	52.1	52.1 ^f^	*50.1*	*55.1(14)*
X_428 K_ (UNEX) ^h^		42(27)		29(57)		11(57)		18(22)		0(6)
X_428 K_ (scan) ^h^		36(10)	42(6)					22(10)		0(10)
R_f_ (tot.) ^g^	4.20									
R_f_ (ind.) ^g^	14.3	7.8	15.8	9.2	15.8	9.8	15.4	9.6	14.2	13.6

^a^ equilibrium r_e_ (QC) and ‘geometrically consistent’ r_h1_ = r_a_ + Δr (GED) internuclear distances, where r_a_ is a vibrationally averaged distance and Δr is a vibrational correction. Parameters marked in *Italic* are for those in boat-shaped cycles. ^b^ M06-2X/aug-cc-pVTZ, parameters obtained by other QC calculations (M06-2X/cc-pVTZ, B3LYP/cc-pVTZ and B3LYP/aug-cc-pVTZ) can be found in ESI, Appendix A. ^c^ Values in parentheses for the GED data are full errors estimated as σ(r_h1_) = [σ_scale_^2^ + (2.5 σ_LS_) ^2^]^½^, where σ_scale_ = 0.002 r and σ_LS_ is a standard deviation in least-squares refinement for internuclear distances and as 3 σ_LS_ for angles, vibration amplitudes and conformers contributions. ^d^ flap angle = 180 − ∠(N-X-X). ^e^ is the dependent parameter. ^f^ is the fixed parameter. ^g^ Disagreement factors include: R_f_ (tot.) for the refined conformers contribution and R_f_ (ind.), assuming that only one individual conformer contributes to the molecular scattering. ^h^ Refined conformers contribution: X_428 K_ (UNEX) evaluated from least-squares refinement of these values along with geometries and vibration amplitudes using the UNEX program and X_428 K_ (scan) from scanning the X values under the refined structures fixed. The 3 σ_LS_ quantities were adopted as error limits in the first case, while the Hamilton’s criterion [6] was applied in the second case.

**Table 4 molecules-28-03933-t004:** Experimental conditions of the synchronous GED/MS experiments.

Parameter	Long	Short
Nozzle-to-plate distance, mm	598	338
Number of recorded films	4	4
Primary electron beam current, μA	1.37	1.49
Exposure time, s	90	100
Accelerating voltage ^a^, kV	74	
Temperature of effusion cell, K	428(5)	
Wavelength of electrons ^b^, Å	0.04403(2)	0.04374(3)
Residual gas pressure, Torr, in		
–diffraction chamber	1.8·10^−6^	1.4·10^−6^
–mass spectrometric block	5.4·10^−7^	5.0·10^−7^
Ionization voltage, V	50	

^a^ Approximate value. ^b^ Accurate wavelengths of electrons were calibrated using diffractions pattern of polycrystalline ZnO.

**Table 5 molecules-28-03933-t005:** EI (U_i_ = 50 V) mass-spectrum of **1** recorded during the GED experiments.

*m/z*	Rel. I, %	Assignment
523	27	[M]^+^
488	25	[M–OF]^+^
439	13	[M–CF_3_O]^+^
390	100	[M–CF_3_SO_2_]^+^
376	14	[M–CF_3_SO_2_N]^+^
365	21	[M–CF_3_SO_2_NC]^+^
322	18	[M–CF_3_SO_2_–CF_3_]^+^
257	20	[M–2CF_3_SO_2_]^+^
189	11	[M–2CF_3_SO_2_–CF_3_]^+^
124	10	[M–3CF_3_SO_2_]^+^

## Data Availability

The data presented in this study are available on request from the corresponding author.

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
