# Peer review of "Gas-Phase Structure of 3,7,9-tris(trifluoromethylsulfonyl)-3,7,9-triazabicyclo[3.3.1]nonane by GED and Theoretical Calculations"

_molecules, 2023, doi:10.3390/molecules28093933_

Round 1
Reviewer 1 Report (Previous Reviewer 3)
The improved version of the manuscript "Gas-phase structure of 3,7,9-tris(trifluoromethylsulfonyl)-3,7,9-triazabicyclo[3.3.1]nonane by GED and Theoretical Calculations" is a much better read and I'm inclined to state that is it interesting enough to warrant publication in Molecules. Authors provide a better introduction and the rationale for the scientific problem to be solved, which was my main critique. The results are OK, but it is always interesting to compare the computational results with the gas-phase experimental results. My only concern in this version is the very bad formatting: many figures are shifted / partially not visible (e.g. Chart 1 and Figure 6) and there are problems in Tables formatting (e.g. in Table S1 some numbers are in italic), therefore I recommend a minor revision.
The language used is OK with some minor problems throughout the text.
Author Response
We thank the Reviewer for his positive review. We tried to correct the format, e.g. in Chart 1, in the Supporting Information, and also corrected some small typos in the text. We also reformatted the list of references according to Molecules rules.
Reviewer 2 Report (Previous Reviewer 1)
The authors of this manuscript present a research article about the conformational composition of a free molecule i.e 3,7,9-tris(trifluoromethylsulfonyl)-3,7,9- 2 triazabicyclo[3.3.1]nonane by using QC calculations and joint GED/MS experiments. The study found that the molecule's conformational composition cannot be described using only one previously detected conformer, and that there are multiple conformers contributing to the equilibrium composition. I reviewed this manuscript earlier and raised a number of concerns. While many of those concerns have been answered, the attitude of the authors does not seem to be of accepting the constructive criticism on their manuscript and in reply of many points, they simply made strange arguments. I have added those comments in a separate heading for reconsideration or at least mentioning those shortcomings explicitly in the manuscript.
Major Points Not Answered satisfactorily:
· The first point has been regretted in a way that is not proper. I can agree if the suggested studies are not of interest to the authors but the studies are not impossible as I suggested “computational studies” for the thermodynamic, elastic and vibrational properties!
· The second point “Limited discussion is done on the relevance and significance of the findings. This manuscript briefly mentions that GED measurements provide unique information on the structure and conformational preferences of the studied molecules. However, the authors do not elaborate on the significance of these findings and how they contribute to the field of chemistry” has been answered in the response to the reviewers but not added to the main manuscript. It must be added to the main manuscript instead of just explaining to me.
· The point which authors did not get was “There is lack of comparison with other experimental techniques like authors only compare their GED results in the gas phase with the single crystal X-ray diffraction experiment. Further the authors explained that their compound is not soluble! Is it insoluble in any solvent? At least these lines should be added to the main manuscript for clarity.
· The point raised by me “It seems that there is limited scope of this study. The authors only focus on one specific molecule, and only analyze conformers and rotamers differing by the conformations of the two piperazine rings and orientation of the CF3 groups. They do not explore other possible variations in the molecule's structure or investigate the effects of external factors, such as temperature or pressure, on the molecule's conformational preferences” has been answered non-professionally. The statement that “Any study is limited” is not necessarily correct and does not answer my concern. So I would like to see this point answered properly in the revised version.
Round 2
Reviewer 2 Report (Previous Reviewer 1)
The revised version seems fine and most of the comments have been addressed. I recommend publication of the manuscript.
This manuscript is a resubmission of an earlier submission. The following is a list of the peer review reports and author responses from that submission.
Round 1
Reviewer 1 Report
The authors of this manuscript present a research article about the conformational composition of a free molecule i.e 3,7,9-tris(trifluoromethylsulfonyl)-3,7,9- 2 triazabicyclo[3.3.1]nonane by using QC calculations and joint GED/MS experiments. The study found that the molecule's conformational composition cannot be described using only one previously detected conformer, and that there are multiple conformers contributing to the equilibrium composition. However, there are several limitations to this study that should be corrected and explained.
Major Points:
· The crystal structure of 3,7,9-tris(tri-fluoromethylsulfonyl)-3,7,9-triazabicyclo[3.3.1]nonane has already find out by the same authors (RSC advances, 2017, 7(62), 38951-38955). So what is the need to study it in gasous phase?. Although the next steps should be to find the thermodynamic, elastic and vibrational properties computationally for its stability, mechanical behaviour, and thermal conductivity, respectively.
· · Limited discussion is done on the relevance and significance of the findings. This manscript briefly mentions that GED measurements provide unique information on the structure and conformational preferences of the studied molecules. However, the authors do not elaborate on the significance of these findings and how they contribute to the field of chemistry.
· · There is lack of comparison with other experimental techniques like authors only compare their GED results with the single crystal X-ray diffraction experiment in the gas phase. However, they do not compare their findings with other experimental techniques, such as NMR spectroscopy in the solution form, which could provide complementary information on the molecule's structure and conformation.
· · This study only examines the molecule in the gas phase, where intermolecular interactions are not present. This may not reflect the molecule's behavior in other phases or in solution, where intermolecular interactions can play a significant role in determining the molecule's conformational composition.
· · This study only examines the molecule at one temperature (428 K), and does not explore how the conformational composition changes with temperature or other conditions. This limits the generalizability of the study's findings.
· It seems that there is limited scope of this study. The authors only focus on one specific molecule, and only analyze conformers and rotamers differing by the conformations of the two piperazine rings and orientation of the CF3 groups. They do not explore other possible variations in the molecule's structure or investigate the effects of external factors, such as temperature or pressure, on the molecule's conformational preferences.
· In table 2, the data for five stable conformers are provided at two different functionals but no conclusion is made. Also, which functional is best in stability of conformer and why? The next question would be if one is better, why use the other one? What was the rationale for selecting these density functionals?
· The data are not explain properly, only tables and figures are added!
Minor Points:
· In figure 2, the red lines shows both the I and III similarly blue lines shows both the I and II while the green only shows I . It seems quite ambigoius and does not match with color rotation in 3d structure. No clear explaintion is provided for this figure.
· The captions of the some figures and tables are not formatted properly.
· There are some typographic mistakes that must be corrected.
· In some areas text is also not formatted properly
Reviewer 2 Report
Manuscript No.: molecules-2309529
Title: Gas-phase structure of 3,7,9-tris(trifluoromethylsulfonyl)-3,7,9 triazabicyclo[3.3.1]nonane by GED and Theoretical Calculations
Comments:
The study of the titled compound is conclusive, well organized and nicely written. Manuscript contains an interesting data that deserve to be published after minor revision. In this paper electronic properties and molecular docking study were carried for the title molecules completely. The manuscript has enough details to consider for publication in Molecules journal after the following comments are properly addressed.
The specific issues are listed below:
· What is the future scope of the present data?
· Why was the level (B3LYP and M06-2X with cc-pVTZ and aug-cc-pVTZ basis sets) chosen for DFT calculations? Justification for the choice of functional and basis set is required.
· The materials nature and properties must include in the manuscript.
· Expand all abbreviations when it first appears in the manuscript
· Is there any relation between the rotation of the trifluoromethylsulfonyl groups with NCI that exists within the molecule.
· Insert physical units at proper places.
· The results and discussion must be revised well and compared to new studies.
· The level of theory needs to be justified. To put the study in its state of the art, the authors must update the Introduction and Results and Discussion sections, by recent works found in the literature. In introduction author need to mention the importance of the NCI effects and DFT calculations with proper references. These Introduction and Results and Discussion sections should be revised with some additions points related to the work with recent references. DOI:10.1016/j.saa.2019.117609, DOI:10.1007/s00894-020-04645-5, DOI :10.1016/j.heliyon.2020.e04724,DOI:10.1016/j.molstruc.2021.130730.
· Occasional grammar mistakes and problems with sentence constructions.
· Conclusion section also needs to be rewritten. Conclusion should contain the significant findings of the work and also discuss on what are the main points which are avenue for the future work.

Reviewer 3 Report
The manuscript "Gas-phase structure of 3,7,9-tris(trifluoromethylsulfonyl)-3,7,9-triazabicyclo[3.3.1]nonane by GED and Theoretical Calculations" delivers exactly what is stated in the title. In general I like the study in the sense that it is well executed from the technical point of view with the calculations supporting the experimental data. I also read with interest the previous study of this group from reference 1 on the same topic. However, while the study from reference 1 shows a new synthetic pathway to obtain a complex compound (although in the current manuscript there is little information why this compound may be interesting from any point of view), the new new study does not really provide, in my opinion, any relevant information. The compound is obviously flexible with multiple minima on the potential energy surface, some of the quite close (in terms of energy) close to each other. Finding the most stable conformer does not really provide any interesting information and authors do not even attempt to validate the goal of the study, beside showing the agreement between the experimental and theoretical data. The PES shown in Figure 2 is what everyone would expect and also shows that the barriers of rotation are so small, that room temperature or higher temperatures the interconversion between the conformers occurs constantly.
As such I believe that this study is not suitable for Molecules, as it does not really provide any relevant data in context of chemistry of this class of compounds.